# Study of Chemical Compositions and Anticancer Effects of *Paris polyphylla* var. *Chinensis* Leaves

**DOI:** 10.3390/molecules27092724

**Published:** 2022-04-23

**Authors:** Feng Su, Lv Ye, Zilin Zhou, An Su, Jinping Gu, Zili Guo, Peixi Zhu, Weike Su

**Affiliations:** 1College of Pharmaceutical Sciences, Zhejiang University of Technology, Hangzhou 310014, China; sufeng@zjut.edu.cn (F.S.); zzl062388@163.com (Z.Z.); jingpingu@zjut.edu.cn (J.G.); guozil@zjut.edu.cn (Z.G.); 2Zhejiang Yangtze Delta Region Pharmaceutical Technology Research Park, Hangzhou 310014, China; 15957109756@163.com; 3Collaborative Innovation Center of Yangtze River Delta Region Green Pharmaceuticals, Zhejiang University of Technology, Hangzhou 310014, China; 4College of Chemical Engineering, Zhejiang University of Technology, Hangzhou 310014, China; ansu@zjut.edu.cn

**Keywords:** *Paris polyphylla* var. *chinensis*, leaves, identification, cytotoxic effect

## Abstract

*Paris polyphylla* var. *chinensis* (Franch.) Hara is a perennial herb belonging to the Trilliaceae family. Ultraperformance liquid chromatography quadrupole time-of-flight mass spectrometry (UPLC/Q–TOF MS) was used to detect the composition of different fractions of *Paris polyphylla* var. *chinensis* leaves. Meanwhile, the extracts of different fractions were evaluated for their cytotoxic activities against four selected human cancer cell lines and one human normal epithelial cell line based on the MTT assay method. Multivariate statistical analysis was performed to screen differential compounds and to analyze the distributions between different fractions. Finally, more than 60 compounds were obtained and identified from the different fractions of *Paris polyphylla* var. *chinensis* leaves, and the chloroform and n−butanol extracts showed significant cytotoxic effects on these four cancer cells. Several compounds were preliminarily identified from different fractions, including 36 steroidal saponins, 11 flavonoids, 10 ceramides, 8 lipids, 6 organic acids, and 8 other compounds. Various compounds were screened out as different chemical components of different fractions, which were considered as a potential substance basis for the cytotoxicity of *Paris polyphylla* var. *chinensis* leaves.

## 1. Introduction

*Paris polyphylla* (PP), which was first recorded by Saoxiu in *Shennong Ben Cao Jing* in China, is a perennial herb belonging to the Trilliaceae family that is distributed in southwest China. The genus *Paris* includes 33 species from all over the world, 26 of which are in China, though only 2 species (*Paris polyphylla* var. *chinensis* (Franch.) Hara and *Paris polyphylla* var. *yunnanensis* (Franch.) Hand.-Mazz.) have been defined by official sources [1]. The rhizoma of PP is called “Chonglou” in traditional Chinese medicine (TCM) and has antipyretic, detoxifying, and swelling and pain relief functions. It is widely used to treat swelling, sore throat, snake bites, throbbing pain, and convulsions [2] and is the key ingredient in many well-known Chinese patent medicines such as the Jidesheng Sheyao tablet, Yunnan Baiyao powder, and Gongxuening capsule [3]. Modern research has shown that the isolate or extract of PP has excellent pharmacological effects such as antitumor [4], antioxidant [5], antibiosis [6], hemostatic [7], and antihelminth [8] effects. The chemical components of PP include steroid saponins, C21 steroidal compounds, flavonoids, phytosterols, polysaccharides, phytoecdysones, and amino acids [9,10]. Among these, steroid saponins play an excellent pharmacological role as the main active ingredients [11,12,13]. 

Because of its excellent medicinal efficacy, PP rhizome is in great demand as a functional medicinal material in the market. However, it takes 7–8 years for PP rhizomes to reach the drug standards. Because of this fact coupled with excessive excavation, the wild resources of PP rhizomes are facing shortages. In contrast, the renewable aboveground parts are discarded in large quantities every year, which may result in a great waste of resources [14]. To broaden the available resources, some scholars have studied the non-medicinal parts of PP, confirming that its aboveground and underground parts have similar chemical compositions and pharmacological activities, especially the leaves [15]. Although many studies have been carried out on the isolation and identification of the chemical constituents in *Paris polyphylla* var. *chinensis* leaves (PPL) [14,15,16,17,18], most have focused on saponins, while little is known about other types of substances and their active ingredients. Therefore, more detailed studies on the phytochemical compositions and pharmacological effects to screen the pharmacoactive substances in the PPL are of great interest.

Solvent type and polarity are known to affect the extract’s quality, quantity, extraction velocity, inhibitory compounds, toxicity, other biological activities, and biosafety [19]. In the current study, solvents of different types and sequential extraction were used for PPL. The components of different fractions were analyzed by ultraperformance liquid chromatography quadrupole time-of-flight mass spectrometry with MS/MS (UPLC/Q−TOF MS/MS) and the potential differences of different fractions were discussed with multivariate statistical analysis. The differences in the cytotoxic of different extracts were studied by MTT method.

## 2. Results and Discussion

### 2.1. Identification of the Chemical Composition from Different Fractions

Substances extracted from various solvents of PPL were analyzed by UPLC/Q−TOF MS/MS and identified by retention time, molecular ion, and main fragmentation information combined with literature accessed via the MassBank database, SciFinder, and DataAnalysis. Most compounds responded in both the positive and negative ion modes but were more sensitive in positive ion mode (Figure 1). The study initially identified 79 compounds from different fractions of PPL, including 36 steroidal saponins, 11 flavonoids, 10 ceramides, 8 lipids, 6 organic acid compounds, and 8 other compounds. Ceramides were first identified in the work on PPL. The results were shown in Table 1.

By comparing the chemical components of the five fractions (petroleum ether fraction, chloroform fraction, ethyl acetate fraction, n−butanol fraction, and water fraction), it was found that there were differences in the substances in the different extracts of PPL. A total of 62 compounds were identified in the n−butanol fraction, mainly steroidal saponins and flavonoids. The compounds in the ethyl acetate fraction were similar to those in the n−butanol fraction, but the content and types were lower. In the petroleum ether fraction, a total of 13 compounds were identified, mainly organic acids and simple alcohols. In the chloroform fraction, 36 compounds were preliminarily identified, among which the content of ceramides was higher and the content of steroidal saponins and flavonoids was lower. The water fraction consisted of small amounts of large-polarity substances.

#### 2.1.1. Identification of Steroidal Saponins

Steroidal saponins are the main active ingredients of PP, and their glucosyl groups are attached to the steroidal aglycones through the hydroxyl groups at C-3 or C-26 [20]. Considering aglycones, four types of steroidal saponins (diosgnin-, pennogenin-, nuatigenin-, and furost-type) are considered to be the main aglycones (Figure 2). Typical sugars are glucose, arabinose, xylose, and rhamnose, which are always located at the C−3 position [21]. Thirty−six steroidal saponins were deduced from different fractions of PPL by comparison with the standards combined with the fragmentation of steroidal saponins [21].

Seven compounds (33, 34, 45–47, 71, and 73) were classified as diosgenin−type saponins. The mass spectra of these compounds indicated the fragmentation ions at *m/z* 415, 397, and 253 [21,22]. Among them, the content of compound 44 was the highest. Compound 44, the molecular ion of which was located at *m/z* 1031.5416 ([M + H]^+^), showed the main fragmentation ions at *m/z* 869.4898 [M + H − Glc]^+^, 577.3749 [869.4898 − 2Rha]^+^, and 415.3219 [577.3749 − Glc]^+^. The fragmentation ions at *m/z* 397.3113 and 271.2063 could be formed by successive losses of H_2_O (18 Da) and C_8_H_16_O_2_ (144 Da). Therefore, compound 44 was identified as pariphyllin A [23]. The fragmentation pathways of compound 44 are proposed in Figure 3. Seven compounds of this type were identified and have been reported previously in PP [23,24].

The main difference between the diosgenin− and pennogenin−type saponins is that pennogenin-type saponins have a hydroxyl group at the C-17 position [23]. Therefore, pennogenin-type saponins are more likely to lose a molecule of water. In PPL, six compounds (31, 38, 39, 42, 44, and 64) were classified as pennogenin-type saponins [12,27,28]. Among them, compound 64 had the highest content. It showed the [M + H − H_2_O]^+^ ion at *m/z* 867.4651. Consequently, it lost two rhamnosyls and a glucosyl to produce the ions at *m/z* 721.4121 [867.4 − Rha]^+^, 575.3556 [721.4 − Rha]^+^, and 413.3036 [575.3 − Glc]^+^. The diagnostic ions at *m/z* 395.2930 and 269.1885 were formed by ion losses of 18 Da (H_2_O) and 144 Da (C_8_H_16_O_2_) at *m/z* 413.3036, respectively. Therefore, compound 64 was identified as chonglouside H [26]. The fragmentation pathways of compound 64 are proposed in Figure 4. 

The nuatigenin−type is characterized by a glucose unit attached to C−26. Diagnostic ions of this type typically appeared at *m/z* 431, 413, and 253, and a glucose unit was lost [23]. Eight compounds (34, 40, 47, 48, 52, 54, 57, and 58) were classified as nuatigenin-type saponins [20,21,28]. The MS2 spectra of protonated compound 40 clearly showed the diagnostic features. Compound 40 showed a protonated molecular ion at *m/z* 1047.5354 [M + H]^+^. After collision-induced dissociation (CID), the primary ions were generated at 885.4841 [1047.5354 − Glc]^+^, 739.4266 [885.4841 − Rha]^+^, 593.3677 [739.4266 − Rha]^+^, 431.3167 [593.3677 − Glc]^+^, and 413.3050 [431.3167 − H_2_O]^+^. The ion at *m/z* 413.3050 produced two fragment ions at *m/z* 395.2944 [413.3050 − H_2_O]^+^ and *m/z* 253.1940 [413.3050 − C_8_H1_4_O_2_]^+^. Based on the fragment ions, compound 41 was identified as 26-*O*-Glc−nuatigenin-3-*O*-rha (1→2)-[rha (1→4)]-glc or its isomer [21]. The fragmentation pathways of compound 40 are proposed in Figure 4. Compound 47 showed similar fragment ions as compound 41, indicating that it might be an isomer of compound 41, and its structural difference lay in the position of the hydroxyl group on the F-ring.

Furost-type saponins are characterized by a split F−ring and a hydroxyl substituent attached at the C-22 position of the aglycone [23]. The [M + H]^+^ ion is typically present in the diosgenin-type but is rarely observed in furost-type saponins because the protonated molecular ions of furost-type saponins are unstable. Eight compounds (28, 30, 32, 36, 37, 50, 51, and 54) were classified as furost−type saponins [26,28]. Among them, compound 36 had the highest content. Compound 36 showed strong [M + H − H_2_O]^+^ at *m/z* 1193.5921, suggesting that the compound was a furost-type saponin. The fragmentation ions at *m/z* 1175.5757, 1013.5306, 867.4716, 721.4159, 575.3583, and 413.3059 resulted from the successive loss of a water molecule (H_2_O), two glucosyls, and three rhamnosyls from the precursor ion at *m/z* 1193.5921. The product ion at *m/z* 413.3059 could lose 144 Da (C_8_H_16_O_2_) to form the fragment ion at *m/z* 269.1905, with further loss of 18 Da (H_2_O) to form the fragmentation ion at *m/z* 251.1793. By comparison with the literature, it was speculated that compound 36 might be the compound Th [28]. The fragmentation pathways of compound 36 are proposed in Figure 4. Compound 30 produced the same fragmentation ions as compound 36; thus, it could be inferred that compound 30 was an isomer of compound 36.

#### 2.1.2. Identification of Flavonoids

As important ingredients in PP, flavonoids exert excellent activities such as antioxidant activity [32,33]. In this work, 11 flavonoids were rapidly detected by UPLC/Q-TOF MS. The structures were preliminarily determined through fragmentation information and the literature. The main structural types were kaempferol, quercetin, luteolin, and isorhamnetin (Figure 5).

Compound 9 showed the [M + H]^+^ ion at *m/z* 641.7133. The fragmentation ions at *m/z* 479.1189 and 317.0662 were produced by elimination of two glucosyl groups. The ion at *m/z* 317.0662 was the aglycon ion of isorhamnetin [34]. Compound 9 was identified as isorhamnetin-3,7-di-*O*-glucoside [28]. Compound 26 showed similar fragment ions to those of compound 9 at *m/z* 479.1202 [M + H]^+^, 317.0670 [M + H − Glc]^+^. Structural differences could be inferred from the lack of glucosyl groups. Therefore, compound 26 was identified as isorhamnetin-3-*O*-glucoside [28].

Compound 8 showed major fragmentation ions at 611.1635 [M + H]^+^, 449.1079 [611.1635 − Glc]^+^, and 287.0550 [449.1079 − Glc]^+^, indicating that the aglycon was luteolin aglycone with two glucoses attached. Compound 18 showed ions at *m/z* 611.1627 [M + H]^+^ and 287.0559 [611.1627 − 2Glc]^+^. The strong aglycon ion at *m/z* 287.0550 in MS2 indicated that the 3−OH group was glycosylated and that the sugar chain was 1→2 connected [34]. Based on the MS and MS2 data, compounds 8 and 18 were assigned as luteolin-3’,7-di-*O*-glucoside and luteolin-3’-*O*-Glc-(l→2)-glucoside, respectively [28].

Compounds 20 and 25 showed ions of [M + H]^+^ at *m/z* 595.1687 and 449.1099, respectively. They showed the same fragmentation ions at *m/z* 287, indicating that their aglycones were both kaempferol glycosides [35]. The mass difference between compounds 20 and 25 was 146 Da, so it could be inferred that there was a difference in rhamnosyl units between these two compounds. Compounds 20 and 26 were estimated to be kaempferol 7-*O*-neohesperidoside and kaempferol 3−glucoside, respectively [35].

#### 2.1.3. Identification of Ceramides

Ceramides are one of the most important sphingolipids formed from sphingosine and fatty acids through amide bonds. It was previously reported that ceramides mainly generated the characteristic product ions of *m/z* 264 and *m/z* 282 in positive ion mode, whereas other sphingosine metabolites provided different fragmentation pathways [36]. The fragments are due to the loss of the *N*−linked fatty acid moiety and one or two water molecules [37]. According to the calibration of high−resolution mass and fragmentation ions, 10 ceramides were identified in the extracts. For example, compound 75 showed a protonated ion at *m/z* 344.3159, indicating an element composition of C_20_H4_1_NO_3_. On the CID, it could form fragmentation ions at *m/z* 300.2896 [M + H − CH_3_CO]^+^, *m/z* 282.2790 [300.2896 − H_2_O]^+^, and *m/z* 264.2693 [282.2790 − H_2_O]^+^. As a result, compound 75 was identified as *D*-erythro-*N*-acetylsphinganine [38].

#### 2.1.4. Identification of Organic Acids

Based on the exact mass data, seven organic acids were identified from the PPL extracts, of which some were inferred as amino acids by comparison with the MassBank database. Amino acids are the basic building blocks of proteins and have pharmacological activities such as regulating the endocrine system, improving immunity and regulating enzyme activity [39]. Yang et al. [40] studied the amino acid content and nutritional values of different parts of PP. The results indicated that the total amount of amino acids in the leaves was abundant and that the leaves therefore had high nutritional value.

### 2.2. Antitumor Activity of Different Fractions In Vitro

All fractions were evaluated for their cytotoxic activities against four human cancer cell lines and one human normal cell line, including A549 (human lung carcinoma), MCF-7 (human breast carcinoma), HepG2 (human hepatocellular carcinomas), A431 (epidermoid carcinoma cell), and HBE (human bronchial epithelial cell). The results are shown in Table 2. Compared with the positive control cisplatin, these fractions were certainly cytotoxic and less toxic to human normal epithelial cells. Among these fractions, the n−butanol and chloroform extracts had the most obvious inhibitory effects against the four cancer cells. The extract of the n−butanol fraction strongly inhibited the proliferation of HepG2 cells, with an IC_50_ value of 0.910 × 10^1^ μg/mL, and chloroform extracts exhibited comparable cytotoxicity against MCF-7 cells, with an IC_50_ value of 1.26 × 10^1^ μg/mL. Other fractions were moderately or weakly cytotoxic, with IC_50_ values ranging from 3.29 × 10^1^ to 8.30 × 10^3^ μg/mL. The results showed that the chemical constituents contained in the chloroform and n−butanol fractions might be effective constituents in inhibiting cancer cells.

### 2.3. Multivariate Statistical Analysis

To differentiate the potential substance basis of PPL inhibiting these cancer cells, metabolites were analyzed using multivariate statistical methods. The obtained multivariate dataset was normalized and then analyzed by SIMCA-P. From the score plot, a clear separation of the five fractions was observed (Figure 6). To identify potential compounds targeting cancer cells, data were analyzed using OPLS-DA. According to the MTT test results, the obtained data were divided into two groups, a high-activity group (chloroform and n−butanol fractions) and a low-activity group (petroleum ether, ethyl acetate, and water fractions). The model described 90.0% of the variation in X (R^2^X = 90.0%), and 98.7% of the variation in Y (class) (R^2^Y = 98.7%), which predicted 98.5% of the variation in Y (Q^2^Y = 98.5%). Hence, the model indicated satisfactory classification among these samples (Figure 6B,6C). The variable importance in the projection (VIP) value was used to screen the variables responsible for bioactivity. Compounds satisfying a VIP value > 1 and a *p*-value < 0.05 are generally considered candidate bioactive compounds. As shown in Table 3, 30 compounds were screened, of which 27 compounds, including steroidal saponins, flavonoids, and ceramides, were highly contained in the high-activity group.

Steroidal saponins have been shown to be the main active ingredient of PP, and the leaves have high content of saponins (roughly 1.36%) [15]. Hu et al. found that steroidal saponins in PPL could induce apoptosis in A549 cells via arrest of the cell cycle in the G0/G1-phase through a mechanism related to Ki-67 and p21 ras protein expression [16]. Furthermore, it could inhibit the proliferation, migration, and invasion of A549 cancer cells by downregulating the matrix metalloproteinases MMP-2 and MMP-9 [41]. The bioactivity of steroidal saponins is closely related to their structural features, and saponins with spirostanol frameworks (including diosgenin, nuatigenin, and pennogenin) have been verified to have medicinal functions. Pariphyllin A, pseudoproto−gracillin and pseudoproto-Pb belong to the diosgenin skeleton. Pariphyllin A showed an inhibitory effect on MCF-7 cells, and the mechanism might be related to the regulation of Bcl−2, Bax, and Caspase-3 [42]; pseudoproto−gracillin and pseudoproto-Pb proved to have potential cytotoxicities against HepG2 cells and A549 cells [43]. Chonglouside 14, a nuatigenin−type saponin obtained from the PPL, displayed cytotoxicity against the HepG2 and HEK293 cell lines with IC_50_ values of 7.0 and 12.9 μM, respectively [44]. Flavonoids are also an important active ingredient and are mainly present in the aerial parts of PP [28]. Kaempferol 7-*O*-neohesperidoside isolated from lychee seeds showed in vitro cytotoxic activity against A549, HepG2, and Hela cell lines with IC_50_ of 0.53, 0.02, and 0.051 μM, respectively [35], while luteolin derivatives could inhibit TGF-β-induced proliferation and invasion of A549 cells [45].

The content of steroidal saponins and flavonoids in the chloroform fraction was lower than in the n-butanol fraction, but the chloroform fraction contained a high proportion of ceramides, which may be the reason for its strong cytotoxicity. Ceramides are important sphingolipids involved in a variety of cellular processes by influencing membrane properties and interacting directly with effector proteins [46]. The ceramides identified from PPL were mainly composed of C2-dihydroceramide, C8-dihydroceramide, sphingosine, and glycosphingolipids. It was reported that C2-ceramide had a strong anticancer effect, as it could inhibit cell proliferation and induce apoptosis by reducing the mitochondrial membrane potential of human colon cancer cell HT-29 [47]. In addition, sphingosine played an important role in basic biological processes such as cell proliferation, differentiation, and tumorigenesis, while glycosphingolipids were essential for the structure and function of cell membranes [48,49].

## 3. Materials and Methods

### 3.1. Plant Materials and Chemicals

The fresh PPL was collected from Hangzhou City (Zhejiang, China) in 01 September, 2020. These samples were identified by Professor Wang Ping from Zhejiang University of Technology. The voucher specimens (No. PPL 20200923) were deposited in Moganshan Campus of Zhejiang University of Technology. The leaves were air−dried naturally at room temperature, crushed, and passed through an 80−mesh sieve. After that, they were stored in a −18 °C freezer until use.

The standards of polyphyllin I (≥98%), II (≥98%), VI (≥98%), and VII (≥95%); dioscin (≥98%); kaempferol 3-*O*-gentiobioside (≥95%); isorhamnetin-3-*O*-glucoside (≥98%); and cisplatin were purchased from Shanghai Yuanye Biotechnology Co. Ltd., (Shanghai, China). Ultrapure water was produced by a Barnstead TII Pure Water System (Waltham, MA, USA). All other analytical−grade chemicals used in this experiment were purchased from Sinopharm Chemical Reagent Co., Ltd. (Shanghai, China). A549 cells, MCF−7 cells, HepG2 cells, A431 cells, and HBE cells were purchased from Hunan Fenghui Biotechnology Co. Ltd. Roswell Park Memorial Institute (RPMI−1640). Dulbecco’s Modified Eagle Medium (DMEM) and fetal bovine serum (FBS) were obtained from Gbico (New York, NY, USA).

### 3.2. Preparation of Samples

Crushed PPL (30 g) were extracted with 75% ethanol three times for 2 h under ultrasonic bath. The supernatant was filtered by decompression–evaporation to give the crude extract. The residue was suspended in distilled water and successively extracted with different solvents, including petroleum ether, chloroform, ethyl acetate, and n−butanol, by liquid–liquid extraction. Each fraction was repeated three times and then pooled and concentrated to dryness using a rotary evaporator and freeze−dryer. The samples were dissolved in methanol and filtered through 0.22 μm membranes as sample solutions for analysis. The quality control (QC) samples were prepared by mixing equal amounts of test solutions and then detected in the same way as samples.

### 3.3. Preparation of Standard Solution

The reference standards, including polyphyllin I, II, VI, and VII; dioscin; kaempferol-3-*O*-gentiobioside; and isorhamnetin-3-*O*-glucoside, were accurately weighed and dissolved in methanol to obtain a stock standard solution at a concentration of 0.5 mg/mL and filtered through 0.22 μm membranes as sample solutions for UPLC analysis.

### 3.4. Operating Conditions of UPLC and MS Analysis

UPLC analysis was performed on an ACQUITY UPLC system (Waters, Milford, MA, USA) equipped with an Acquity UPLC HSS T3 column (100 × 2.1 mm, 1.8 μm). The column temperature was set at 35 °C, the injection volume as 2 μL, and the flow rate as 0.2 mL/min. The chromatogram was collected at the wavelength of 203 nm by the VWD detector. The mobile phases were acetonitrile (A) and 0.1% formic acid (B). The gradient elution was: 0–5 min, 5–15% A; 5–18 min, 15–25% A; 18–25 min, 25–35% A; 25–35 min, 35–50% A; 35–40 min, 50–60% A; 40–47 min, 60–98% A; 47–55 min, 98% A; 55–55.1 min, 98−5% A; 55.1–60 min, 5% A.

A microTOF-compact mass spectrometer (Bruker Daltonics, Bremen, Germany) using composite electrospray ionization (ESI) in both positive and negative ion modes was utilized, and the optimized conditions were as follows: capillary voltage was 4500 V; dry gas (N_2_) flow rate was 4 L/min; nebulizer gas (N_2_) pressure was 2.2 psi; dry gas temperature was 220 °C. The scan range was 50 to 2000 Da. Sodium formate calibration solution with a concentration of 10 mM was used as calibration solution for each needle sample. Each batch of samples was tested 6 times in parallel, and every 6 analyses included an inspection of QC samples.

### 3.5. Cytotoxic Assay

Four human cancer cell lines (A549, MCF-7, HepG2, and A431) and one human normal epithelial cell line were used for cytotoxicity assay. All the cells were cultured in RPMI−1640 or DMEM medium, supplemented with 10% fetal bovine serum, and cultured in a cell incubator at 37 °C and 5% CO_2_. A suspension of the cells at logarithmic phase was seeded in 96-well plates at a density of 5000 cells per well (100 μL of medium per well) and cultured for 24 h for cell adhesion. In the experimental group, extracts from five different parts were dissolved in phosphate−buffered saline (PBS) and diluted with medium to various concentrations. Each cell line was exposed to the test samples at various concentrations for 48 h, with cisplatin as positive control. The wells of the control group medium contained 100 μL medium, while wells without cells served as blank control. After 48 h incubation, the supernatant was discarded, and 20 μL of MTT solution (5 g/L) was added to each well and incubated for 4 h at 37 °C. Subsequently, the supernatant was carefully removed by suction, and 150 μL DMSO was added to each well to dissolve the crystals. The absorbance of each solution was read at 570 nm against blank control with the reference wavelength at 620 nm in a microplate reader, and the IC_50_ values were calculated based on the results.

### 3.6. Data Analysis

The MS/MS data were collected by the Hystar 3.2 software, and the collected data of six batches were internally corrected in enhanced mode by Bruker Daltonics DataAnalysis 4.4 software with sodium formate calibration solution. The processed data were converted into Analysis Base File (ABF) format using AbfConverter (Version 4.0.0) and then imported into MS-Dial (Version 4.16) for preprocessing including peak collection, peak discrimination, deconvolution, filtering, peak alignment, and normalization. After that, a data matrix containing average retention time, average retention index, EI spectrum, sample name, and peak area was obtained. Then, the three−dimensional data matrix was introduced into SIMCA 14.1 (Umetrics, Sweden) for PCA and OPLS-DA analysis. All of the variables were normalized by Par (Pareto−scaled) before stoichiometric analysis.

## 4. Conclusions

In the current study, UPLC/Q−TOF MS/MS was used to qualitatively analyze the chemical composition of different fractions in PPL alcohol extracts, and 79 compounds were finally identified. The results of an MTT test and multivariate statistical analysis showed that the chloroform and n−butanol fractions displayed interesting cytotoxicities against four human cancer cell lines, with IC_50_ values ranging from 0.910 × 10^1^ to 1.68 × 10^1^ μg/mL, and were less toxic to a human normal epithelial cell line. The proportions and contents of steroidal saponins, flavonoids, and ceramides in the chloroform and n−butanol fractions were higher, which might be the reason for these fractions’ high inhibitory effect on cancer cells. The basis for the potential anticancer substances in PPL was preliminarily determined, which provided a theoretical basis for the extraction of active ingredients and also laid a foundation for a follow-up study of the specific mechanism of the inhibitory effect of the active ingredients of PPL on A549 cancer cells. Accordingly, the experimental results proved that PPL had potential development and utilization prospects, which is of great value to the development of PPL products.

## Figures and Tables

**Figure 1 molecules-27-02724-f001:**
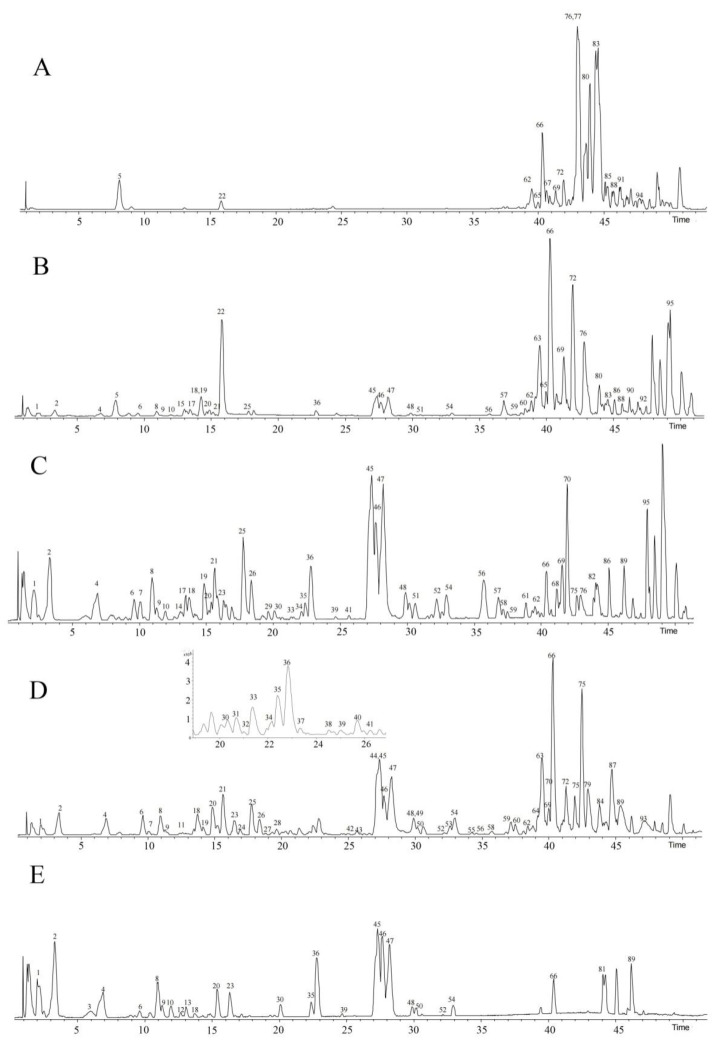
Base peak chromatogram (BPC) of different fractions from the *Paris polyphylla* var. *chinensis* leaves (PPL) in positive ion mode. (**A**). Petroleum ether fraction; (**B**). chloroform fraction; (**C**). ethyl acetate fraction; (**D**). n−butanol fraction; (**E**). water fraction.

**Figure 2 molecules-27-02724-f002:**
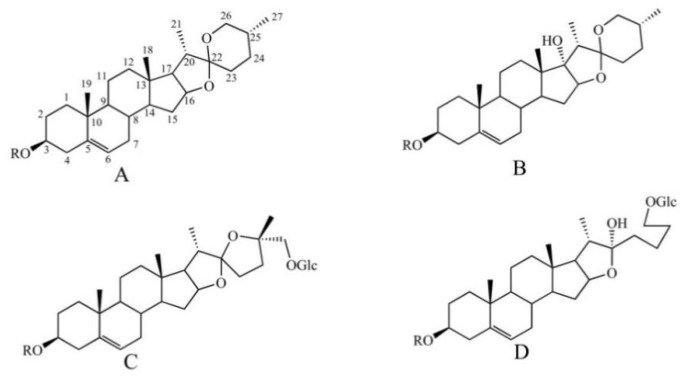
The steroidal saponin skeleton identified from different parts of PPL in positive ion mode. A. Diosgnin-type; B. pennogenin-type; C. nuatigenin-type; D. furost-type.

**Figure 3 molecules-27-02724-f003:**
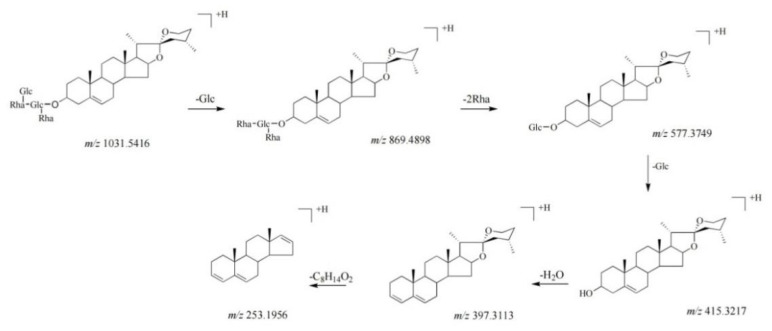
The fragmentation pathways of compound 44 in positive ion mode [21,25,26].

**Figure 4 molecules-27-02724-f004:**
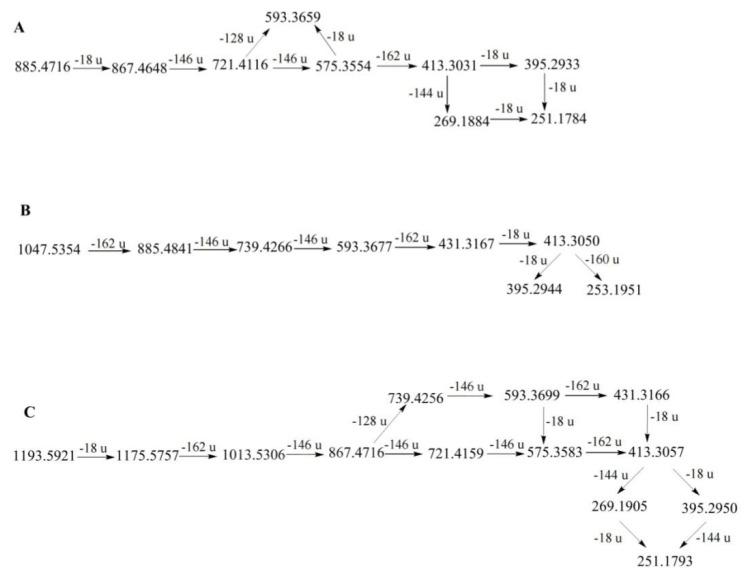
The fragmentation pathways of compounds 64 (**A**), 40 (**B**), and 36 (**C**) in positive ion mode [23,26,29,30,31].

**Figure 5 molecules-27-02724-f005:**
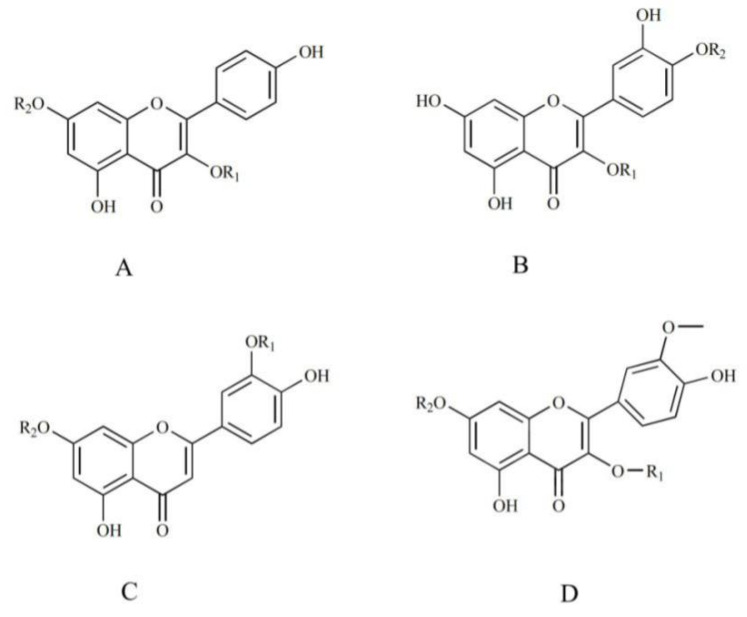
The flavonoid skeletons identified from different parts of PPL in positive ion mode. (**A**) Kaempferol-type; (**B**). quercetin-type; (**C**) luteolin-type; (**D**) isorhamnetin-type.

**Figure 6 molecules-27-02724-f006:**
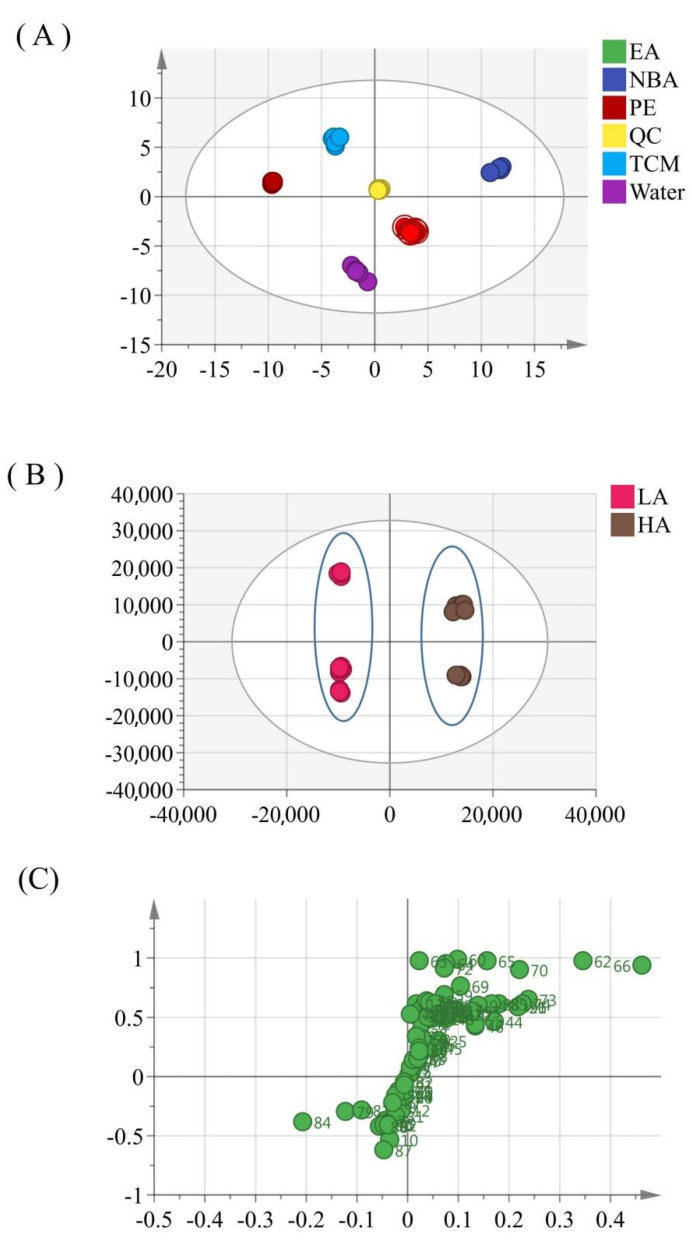
Score plots from principal component analysis (**A**), orthogonal partial least squares discriminant analysis (**B**), and loading plot models (**C**) in different layers from PPL.

**Table 1 molecules-27-02724-t001:** Compounds identified in the extracts of different fractions from PPL by UPLC/Q−TOF MS/MS.

No	tR/Min	[M + H]^+^ *m/z*	Molecular Formula	Error (ppm)	ESI/Q–TOF MS/MS Fragments	Tentative Identification	Existing PART
1	2.2	294.1547	C_12_H_23_NO_7_	0.3	276.1440 (100), 258.1336 (50.4), 230.1386 (39.5), 132.1020 (15.5), 248.1488 (10.9)	N-fructosyl isoleucine	TCM, EA, NBA, Water
2	3.4	328.1392	C_15_H_21_NO_7_	−0.3	310.1283 (100), 292.1178 (57.3), 264.1228 (33.2), 166.0861 (28.3)	N-fructosyl phenylalanine	TCM, EA, NBA, Water
3	6.0	367.1501	C_17_H_22_N_2_O_7_	−0.5	229.0966 (100), 349.1386 (81.4), 188.0699 (74.8), 332.1128 (61.5), 276.1229 (38.3), 258.1122 (21.3), 350.1417 (14.9)	N.A.	Water
4	6.8	205.0969	C_11_H_12_N_2_O_2_	1.3	188.0706 (100), 146.0605 (82.0), 144.0813 (17.1), 159.0918 (11.9)	Tryptophan	TCM, EA, NBA, Water
5	8.2	192.0923	C_9_H_9_N_3_O_2_	0.5	160.0502 (100), 192.0760 (29.1)	Carbendazim	PE, TCM, EA
6	9.6	217.0970	C_12_H_12_N_2_O_2_	0.9	144.0811 (100), 145.0842 (10.0)	Lycoperodine I	TCM, EA, NBA, Water
7	10.1	169.0502	C_8_H_8_O_4_	−3.7	125.0603 (100), 151.0389 (83.5), 269.0499 (30.0)	2,4,6-trihydroxyacetophenone	EA, NBA
8	11.0	611.1634	C_27_H_30_O_16_	−4.5	287.0550 (100), 449.1079 (26.0), 288.0583 (14.9)	Luteolin-3’,7-di-*O*-glucoside	TCM, EA, NBA, Water
9	11.4	641.1738	C_28_H_32_O_17_	−4.0	317.0664 (100), 479.1193 (25.8), 318.0690 (13.6)	Isorhamnetin-3,7-di-*O*-glucoside	TCM, EA, NBA, Water
10	12.1	697.1630	C_30_H_32_O_19_	−2.9	287.0548 (100), 449.1086 (52.9), 288.0582 (14.4), 127.0401 (11.6), 450.1119 (11.3)	3-[[6-*O*-(2-Carboxyacetyl)-4-*O*-hexopyranosyl-β-d-glucopyranosyl]oxy]-5,7-dihydroxy-2-(4-hydroxyphenyl)-4H-1-benzopyran-4-one	TCM, EA, EA, Water
11	12.4	627.1555	C_27_H_30_O_17_	0.1	303.0497 (100), 304.0528 (13.2)	Quercetin-3,4’-*O*-di-beta-glucoside	NBA
12	12.7	551.2689	C_25_H_42_O_13_	1.6	209.1533 (100), 149.0959 (68.9), 227.1637 (55.2), 191.1426 (25.6)	Tricalysionoside A	Water
13	13.1	252.0865	C_12_H_13_NO_5_	0.5	206.0809 (100), 120.0806 (93.5), 188.0703 (48.1), 146.0597 (15.9), 207.0842 (10.0)	N.A.	Water
14	13.1	165.0553	C_9_H_8_O_3_	−3.9	147.0446 (100), 148.0418 (9.3), 123.0919 (2.4)	*O*-coumaric acid	EA
15	13.2	227.1649	C_13_H_22_O_3_	−2.7	125.0970 (100), 149.0970 (87.7), 123.0818 (63.7), 153.0918 (43.1), 209.1541 (33.7), 191.1437 (23.1)	4-(1-Hydroxy-4-keto-2,6,6-trimethyl-2-cyclohexen-1-yl)-butan-2-ol	TCM
16	13.5	611.1605	C_27_H_30_O_16_	0.3	303.1493 (100), 304.0525 (13.3)	Rutin	NBA
17	13.5	225.1499	C_13_H_20_O_3_	−5.3	161.1335 (100), 179.1440 (26.0), 121.1024 (25.1), 162.1369 (10.3)	2-(6-methyl-7-oxooctyl)-2H-furan-5-one	TCM, EA
18	13.8	611.1627	C_27_H_30_O_16_	−3.3	287.0559 (100), 288.0586 (14.7)	Luteolin-3’-*O*-Glc-(l→2)-glucoside	TCM, EA, NBA, Water
19	14.2	641.1742	C_28_H_30_O_17_	−4.7	317.0672 (100), 318.0700 (15.2)	Isorhamnetin-3-*O*-glc-(l→2)-gal	TCM, EA, NBA
20	14.9	595.1685	C_27_H_30_O_15_	−4.7	287.0558 (100), 288.0595 (14.6)	Kaempferol 7-*O*-neohesperidoside	TCM, EA, NBA, Water
21	15.6	481.3170	C_27_H_44_O_7_	−2.2	445.2956 (100), 371.2228 (39.0), 427, 2857 (33.2), 165.1281 (26.7), 409.2732 (11.6), 428.2894 (10.1)	Crustecdysone	TCM, EA, NBA
22	15.9	197.1167	C_11_H_16_O_3_	0.5	179.1063 (100), 197.1163 (49.7), 135.1171 (34.4), 133.1013 (33.1), 161.0955 (26.0)	Loliolide	PE, TCM
23	16.4	1225.5500	C_56_H_88_O_29_	−1.3	621.3268 (100), 553.3008 (70.0), 459.2736 (59.6), 391.2473 (47.8), 373.2366 (33.2), 622.3300 (29.2), 441.2841 (26.3), 767.3855 (24.9), 477.2841 (24.4), 605.3317 (24.3), 699.3578 (22.7), 639.3374 (20.8), 309.1176 (19.6), 783.3797 (16.0), 929.4381 (7.5)	Parisyunnanoside G	EA, NBA, Water
24	16.5	481.3164	C_27_H_44_O_7_	−0.9	427.2852 (100), 303.1957 (69.5), 143.1070 (66.3), 125.0964 (48.2), 285.1850 (44.2), 409.2744 (42.0), 428.2883 (25.6), 301.1802 (22.1)	Crustecdysone or its isomer	NBA
25	17.8	449.1093	C_21_H_20_O_11_	−3.2	287.0555 (100), 288.0589 (14.9)	Kaempferol 3-Glucoside	TCM, EA, NBA
26	18.4	479.1189	C_22_H_22_O_12_	−1.0	317.0655 (100), 318.0688 (16.7)	Isorhamnetin-3-*O*-glucoside	EA, NBA
27	18.9	1063.5301	C_51_H_82_O_23_	1.8	293.1236 (100), 427.2850 (78.0), 147.0653 (42.6), 445.2956 (40.6), 239.0920 (36.6), 257.1024 (32.3), 409.2743 (29.6), 461.2898 (22.7), 309.1185 (21.8), 589.3380 (21.3), 607.3489 (16.8), 735.3948 (14.3)	N.A.	NBA
28	19.7	901.4769	C_45_H_72_O_18_	2.5	269.1894 (100), 287.1998 (69.7), 147.0647 (67.1), 441.2631 (57.4), 427.2829 (55.7), 595.3133 (46.6), 739.4247 (45.9), 901.4781 (39.0), 721.4162 (33.9), 409.2742 (30.9)	26-*O*-glc-furost-5-ene-3β,22α,26-triol-3-*O*-rha-(1→2)-glc or its isomer	NBA
29	19.7	1047.5328	C_51_H_82_O_22_	3.2	413.3053 (100), 147.0647 (70.1), 395.2945 (65.8), 129.0549 (51.8), 1047.5367 (36.1), 269.1890 (31.8), 431.3149 (30.7), 281.2269 (29.9), 377.2836 (23.4), 885.4798 (18.0), 849.4619 (13.7), 1063.5284 (4.2)	26-*O*-Glc-nuatigenin-5-ene-3β,17-diol-3-*O*-rha (1→2) -[rha (1→4)]-glc	EA, NBA
30	20.1	1193.5995	C_57_H_94_O_27_	−3.8	413.3053 (100), 1193.5980 (52.8), 431.3162 (50.4), 293.1230 (49.1), 395.2947 (37.8), 147.0658 (29.4), 414.3086 (22.5), 593.3688 (16.9), 739.4295 (5.9)	Th or its isomer	EA, NBA, Water
31	20.7	901.4815	C_45_H_72_O_18_	−2.6	269.1906 (100), 287.2003 (72.0), 147.0661 (27.9), 413.3043 (26.6), 595.3112 (22.1), 901.4785 (21.6), 739.4256 (21.1), 431.3165 (12.0)	(25*S*)-spirost-5-ene-3β,25-diol-3-*O*-glc (1→3)-[rha (1→2)]-glc	EA, NBA
32	21.0	1063.5289	C_51_H_82_O_23_	2.9	445.2944 (100), 293.1227 (90.2), 147.0649 (53.6), 239.0912 (41.3), 607.3480 (39.9), 427.2839 (38.8), 753.4068 (34.1), 257.1014 (32.9), 129.0552 (29.5), 275.1111 (25.2), 446.2976 (23.6), 271.2045 (21.6)	26-*O*-β-d-Glc-3β,12,22,26-tetrahydroxyfurost-5-ene-22,25-epoxy-3-*O*-Rha-(1→2)-[Rha-(1→4)-Rha-(1→4)]-β-D-Glc	NBA
33	21.4	755.4245	C_39_H_62_O_14_	−4.3	269.1908 (100), 449.2554 (65.4), 593.3715 (38.3), 287.2021 (37.5), 755.4265 (30.0), 270.1949 (18.5), 413.3075 (16.2), 251.1811 (15.4), 594.3748 (15.3), 450.2583 (13.3)	(25*S*)-spirost-5-ene-3-*O*-glc-(1→2)-glc	EA, NBA
34	22.0	933.4671	C_45_H_72_O_20_	2.0	445.2948 (100), 147.0652 (91.1), 427.2846 (65.9), 129.0549 (41.5), 309.1183 (36.3), 271.2054 (30.3), 253.1942 (20.3), 163.0605 (12.4), 607.3457 (7.0)	(23*S*,24*S*)-spirost-5-ene-1β,3β,23,24-tetrol-1-*O*-rha(1→2)-glc 24-*O*-gal	EA, NBA
35	22.4	1047.5393	C_51_H_82_O_22_	−2.2	413.3048 (100), 867.4745 (75.1), 147.0655 (48.6), 395.2945 (35.5), 129.0552 (29.0), 868.4683 (28.5), 431.3157 (26.7), 293.1235 (26.3), 414.3080 (24.5), 181.1224 (20.7), 309.1170 (12.8), 239.0913 (10.5)	27-*O*-glc-(25*R*)-spirost-5-ene-3β,27-diol-3-*O*-rha(1→4)-[rha(1→2)]-glc	EA, NBA, Water
36	22.9	1193.6061	C_57_H_94_O_27_	−5.0	413.3065 (100), 293.1242 (88.7), 431.3173 (42.0), 147.0660 (36.5), 257.1027 (25.0), 593.3709 (16.4), 721.4181 (6.2), 1013.5360 (6.3)	Th	TCM, EA, NBA, Water
37	23.3	917.4738	C_45_H_72_O_19_	0.3	269.1890 (100), 737.4085 (71.6), 287.2001 (64.7), 147.0645 (48.2), 755.4207 (48.0), 411.2890 (47.8), 429.2990 (46.3), 595.3108 (21.6), 756.4239 (21.5)	26-*O*-β-d-Glc-3β,22,26-trihydroxyfurost-5-ene-3-*O*-Rha-glc	NBA
38	24.5	917.4745	C_45_H_72_O_19_	−0.5	269.1903 (100), 755.4188 (82.8), 287.1999 (70.4), 429.2998 (70.4), 447.3101 (47.0), 411.2904 (38.6), 756.4218 (29.1), 609.3613 (25.5), 595.3117 (18.4), 251.1799 (16.2)	(25*R*)-spirost-5-ene-3β,17α,27-triol-3-*O*-glc(1→3)-[rha(1→2)]-glc or its isomer	NBA
39	24.7	1209.5925	C_57_H_92_O_27_	−2.2	591.3527 (100), 429.3001 (85.0), 293.1231 (67.0), 737.4091 (54.8), 197.1168 (37.0), 592.3553 (30.1), 147.0651 (26.3), 573.3424 (26.1), 411.2886 (25.0), 447.3101 (21.6), 430.3039 (20.5), 899.4631 (18.4)	Parisverticoside C	EA, NBA, Water
40	25.0	593.3695	C_33_H_52_O_9_	−1.8	287.2008 (100), 431.3165 (52.1), 593.3685 (21.8), 269.1901 (20.3), 288.2037 (17.6), 432.3180 (12.7), 167.1070 (10.3)	Chonglouoside SL-1	NBA
41	25.7	1047.5370	C_51_H_82_O_22_	0.1	431.3143 (100), 885.4838 (70.6), 148.0652 (59.8), 413.3040 (56.2), 129.0547 (36.0), 271.2046 (32.6), 593.3659 (31.1), 253.1947 (25.0), 739.4249 (17.9), 395.2919 (13.1), 447.3102 (11.7)	26-*O*-Glc-nuatigenin-3-*O*-rha (1→2) -[rha (1→4)]-glc	EA, NBA
42	26.0	901.4779	C_45_H_72_O_18_	−3.1	129.0547 (100), 147.0652 (97.7), 411.2902 (69.2), 393.2795 (62.5), 281.2270 (42.2), 269.1905 (41.2), 429.3003 (37.2), 251.1793 (36.5), 901.4808 (34.5), 557.3534 (9.8), 739.4267 (9.2)	(25*R*)-spirost-5-ene-3β,17β,27-triol-3-*O*-rha(1→4)-[rha(1→2)]-glc or its isomer	NBA
43	26.2	1047.5347	C_51_H_82_O_22_	2.2	447.3119 (100), 147.0654 (69.0), 293.1229 (62.8), 429.3003 (55.2), 411.2911 (55.2), 129.0554 (49.9), 239.0918 (39.0), 593.3685 (37.8), 755.4194 (37.0), 609.3632 (27.8), 275.1123 (24.2), 257.1023 (24.1), 393.2802 (20.5), 885.4821 (7.0)	N.A.	NBA
44	27.4	739.4271	C_39_H_62_O_13_	−1.1	577.3748 (100), 433.2591 (73.9), 253.1952 (69.5), 271.2058 (38.7), 739.4270 (31.5), 578.3780 (29.1), 167.1065 (20.5), 397.3122 (9.3)	Polyphyllin VI	NBA, Water
45	27.4	1031.5461	C_51_H_82_O_21_	−3.8	415.3228 (100), 869.4946 (99.0), 1031.5479 (92.1), 147.0663 (83.6), 271.2068 (70.0), 253.1964 (61.4), 129.0557 (52.8), 167.1076 (43.5), 870.4980 (41.8), 725.3785 (26.5), 577.3760 (11.2)	Pariphyllin A	TCM, EA, NBA, Water
46	27.7	1177.6053	C_57_H_92_O_25_	−4.5	415.3224 (100), 1177.6060 (81.2), 293.1240 (60.1), 577.3763 (40.2), 147.0661 (30.0), 271.2066 (23.8), 416.3260 (23.3), 397.3114 (19.9), 723.4351 (14.3), 869.4926 (5.0)	Pseudoproto-Pb	TCM, EA, NBA, Water
47	28.2	1047.5436	C_51_H_82_O_22_	−4.3	415.3230 (100), 147.0665 (81.4), 271.2070 (77.9), 1047.5440 (76.7), 885.4900 (60.1), 253.1964 (55.4), 397.3124 (40.9), 129.0559 (39.9), 577.3772 (17.4), 741.3739 (11.3)	Pseudoproto-gracillin	TCM, EA, NBA, Water
48	29.9	1047.5408	C_51_H_82_O_22_	−3.6	431.3168 (100), 885.4877 (60.4), 147.0662 (48.8), 413.3067 (48.1), 593.3703 (35.9), 129.0557 (33.2), 271.2065 (30.4), 739.4277 (25.3), 432.3203 (23.9), 309.1182 (12.7)	26-*O*-Glc-nuatigenin-3-*O*-rha (1→2) -[rha (1→4)]-glc or its isomer	TCM, EA, NBA, Water
49	29.9	901.4798	C_45_H_72_O_18_	−0.8	739.4271 (100), 593.3709 (74.4)147.0654 (57.9), 253.1951 (51.3), 271.2062 (43.9), 129.00546 (43.3), 431.3155 (40.8), 413.3056 (37.6), 579.3176 (24.2), 395.2963 (19.7), 575.3583 (16.1)	27-*O*-glc-(25*R*)-spirost-5-ene-3β,27-diol-3-*O*-rha(1→4)-glc	NBA
50	30.2	1193.6005	C_57_H_94_O_27_	−4.7	431.3172 (100), 593.3710 (53.5), 293.1240 (44.3), 413.3068 (33.0), 432.3208 (22.5), 739.4322 (22.5), 129.0556 (13.6), 885.4888 (10.0)	Chonglouoside SL-14	TCM, EA, NBA, Water
51	30.7	1063.5366	C_51_H_84_O_24_	−4.3	431.3172 (100), 413.3061 (51.5), 593.3703 (43.6), 147.0661 (40.0), 432.3206 (24.0), 309.1185 (23.6), 575.3598 (23.1), 129.0557 (19.8), 755.4256 (13.6), 414.3098 (13.2), 271.2067 (11.0), 901.4828 (6.3)	26-*O*-glc-furost-5-ene-3β,12α,22α,26-trihydroxy-3-*O*-glc-(1→3)-[rha-(1→2)]-glc	TCM, EA, NBA, Water
52	32.2	1209.5821	C_57_H_92_O_27_	4.7	431.3149 (100), 593.3679 (32.3), 413.3048 (20.0), 293.1221 (16.7), 739.4238 (15.4), 309.1170 (9.4), 885.4810 (4.0)	N.A.	EA, NBA, Water
53	32.6	1047.5404	C_51_H_82_O_22_	−3.2	885.4870 (100), 431.3160 (83.7), 886.4898 (45.7), 867.4745 (43.8), 147.0658 (39.9), 413.3054 (36.9), 253.1946 (24.9), 271.2061 (24.4), 129.0550 (22.6), 868.4754 (16.2), 725.3727 (13.4), 593.3687 (13.4), 739.4339 (11.6), 239.0934 (10.0)	27-*O*-glc-(25*R*)-spirost-5-ene-3β,27-diol-3-*O*-rha(1→4)-[rha(1→2)]-glc or its isomer	NBA
54	33.0	1193.5962	C_57_H_92_O_26_	−1.1	431.3158 (100), 593.3685 (27.2), 413.3051 (24.5), 432.3192 (24.4), 293.1231 (23.7), 739.4278 (21.0), 885.4858 (14.3), 147.0655 (10.3), 886.4903 (6.3)	Chonglouoside SL-14 or its isomer	TCM, EA, NBA, Water
55	34.3	885.4836	C_45_H_72_O_17_	0.7	147.0651 (100), 431.3159 (83.6), 129.0548 (80.2), 413.3053 (67.6), 253.1952 (66.8), 885.4835 (50.4), 271.2062 (48.8), 293.1229 (37.0), 593.3691 (19.9), 739.4281 (13.2)	(25*S*)-isonuatigenin-3-*O*-rha-(1→4)-[rha-(1→2)]-glc	NBA
56	35.8	769.2314	C_45_H_36_O_12_	0.7	645.1775 (100), 375.0871 (40.1), 646.1813 (32.7), 389.1025 (30.8), 137.0605 (25.4), 521.1247 (24.7), 513.1553 (22.1), 259.0979 (16.0), 769.2319 (11.2), 377.1014 (11.2)	N.A.	TCM, EA, NBA
57	36.9	657.4610	C_36_H_64_O_10_	−5.0	173.1182 (100), 155.1077 (95.0), 275.2016 (44.5), 293.2122 (24.2)	N.A.	TCM, EA
58	37.2	1031.5364	C_51_H_82_O_21_	5.0	885.4802 (100), 593.3672 (49.3), 1031.5364 (31.2), 739.4238 (27.5), 431.3147 (26.1), 293.1225 (9.3), 413.3041 (6.0)	Parisyunnanside C	EA, NBA
59	37.5	901.4815	C_45_H_72_O_18_	−2.6	431.3168 (100), 147.0660 (89.0), 413.3064 (76.8), 129.0556 (43.6), 309.1185 (26.7), 432.3209 (24.7), 271.2062 (23.4), 414.3080 (20.9), 253.1957 (18.5), 273.0974 (13.6), 163.0608 (12.4), 145.0500 (11.8)	(25*S*)-spirost-5-ene-3β,25-diol-3-*O*-rha (1→2)-[glc (1→3)] -glc	TCM, EA, NBA
60	38.4	494.3339	C_24_H_47_NO_9_	−3.1	332.2801 (100), 494.3340 (80.8), 495.3372 (19.8), 333.2835 (17.4)	HexCer t18:0	TCM, NBA
61	38.9	683.4726	C_38_H_66_O_10_	0.3	353.2304 (100), 354.2336 (18.6)	n.a.	EA
62	39.5	496.3487	C_24_H_49_NO_9_	−0.7	334.2950 (100), 496.3483 (92.3), 497.3514 (25.1), 335.2981 (18.6)	Hydroxyl 1-*O*-(β-d-glc)-(2*S*, 3*S*)-2-acetamide-4 (E)-octadecane-1,3,6-triol	PE, TCM, EA, NBA
63	39.8	1031.5388	C_51_H_82_O_21_	3.2	413.3034 (100), 293.1219 (72.9), 721.4122 (52.5), 395.2929 (49.6), 575.3558 (40.0), 147.0644 (29.4), 239.0901 (29.2), 257.1009 (24.5), 431.3138 (13.6), 867.4667 (4.4)	Polyphyllin VII	TCM, EA, NBA
64	40.0	885.4798	C_45_H_72_O_17_	5.0	413.3036 (100), 395.2930 (71.9), 147.0645 (45.5), 129.0542 (29.8), 293.1214 (29.7), 575.3556 (22.8), 239.0897 (19.3), 396.2958 (16.2), 431.3143 (15.2), 275.1113 (12.8), 721.4121 (7.7)	Chonglouside H	NBA
65	40.0	334.2965	C_18_H_39_NO_4_	−3.8	316.2849 (100), 334.2963 (53.3), 298.2745 (30.4), 280.2635 (18.7), 317.2887 (17.5), 335.2985 (0.5)	3,3’-(dodecylimino)bispropane-1,2-diol	PE, TCM, EA
66	40.3	480.3542	C_24_H_49_NO_8_	−1.1	318.3004 (100), 480.3538 (56.0), 300.2897 (11.1)	1-*O*-(β-d-glc)-(2*S*,3*S*)-2-acetamide-4 (E)-octadecane-1,3,6-triol	PE, TCM, EA, NBA, Water
67	40.7	275.2006	C_18_H_26_O_2_	0.0	173.1328 (100), 137.0969 (78.9), 257.1398 (27.0), 275.2002 (14.1), 239.1792 (11.0)	Nandrolone	PE
68	41.2	392.3012	C_20_H_41_NO_6_	−1.3	279.2319 (100), 356.2796 (62.4), 374.2898 (35.4), 338.2690 (20.7), 280.2345 (15.9), 144.0663 (15.4), 357.2822 (12.3)	N-tetradecyl-d-gluconamide	EA
69	41.3	318.3004	C_18_H_39_NO_3_	−0.1	318.2999 (100), 300.2893 (67.6), 282.2788 (63.7), 301.2924 (12.3)	Phytosphingosine	PE, TCM, EA, NBA
70	41.6	506.3698	C_26_H_51_NO_8_	−2.0	344.3158 (100), 506.3693 (68.3), 300.2897 (32.7), 345.3192 (20.6), 507.3729 (18.2), 282.2792 (11.0)	N-[1-[(β-d-Galactopyranosyloxy)methyl]-2-hydroxyheptadecyl]acetamide	EA, NBA
71	41.7	1015.5420	C_51_H_82_O_20_	4.2	293.1222 (100), 415.3197 (91.7), 397.3089 (81.0), 147.0647 (75.6), 239.0904 (57.4), 129.0541 (50.6), 257.1011 (39.6), 723.4291 (25.8), 577.3709 (21.8), 398.3118 (18.8), 309.1166 (16.3), 869.4854 (5.8)	Polyphyllin II	EA, NBA
72	42.0	376.3059	C_20_H_41_NO_5_	−0.2	340.2854 (100), 280.2635 (58.1), 358.2951 (34.2), 263.2377 (28.2), 322.2738 (14.7), 262.2528 (13.5)	1-Deoxy-1-(tetradecylamino)-d-fructose	PE, TCM, EA, NBA
73	42.2	869.4864	C_45_H_72_O_16_	3.4	869.4865 (100), 147.0648 (97.2), 253.1944 (83.4), 129.0545 (75.4), 271.2047 (46.8), 293.1226 (35.2), 397.3090 (34.8), 415.3199 (29.4), 725.3731 (22.5), 239.0905 (19.2), 577.3711 (5.2)	Dioscin	NBA
74	42.5	534.3999	C_28_H_55_NO_8_	0.2	434.3999 (100), 372.3469 (82.0), 300.2892 (49.5), 535.4034 (31.4), 462.3423 (26.5), 373.3504 (19.6), 282.2786 (16.8), 222.0969 (16.8)	N-[1-[(β-d-Galactopyranosyloxy)methyl]-2-hydroxyheptadecyl]butanamide	NBA
75	42.7	344.3159	C_20_H_41_NO_3_	0.0	300.2900 (100), 282.2795 (70.4), 344.3162 (51.4), 301.2930 (17.5), 283.2827 (12.1)	d-erythro-N-Acetylsphinganine	EA, NBA
76	43.0	346.3312	C_20_H_43_NO_3_	1.2	346.3311 (100), 328.3206 (54.1), 310.3101 (43.3), 347.3343 (22.5), 329.3239 (11.3)	1-[Bis(2-hydroxyethyl)amino]-2-hexadecanol	PE, TCM, EA, NBA
77	43.0	277.2163	C_18_H_28_O_2_	−0.1	121.1023 (100), 135.1177 (94.4), 277.2162 (65.6), 149.1330 (34.2), 133.1021 (23.5), 147.1174 (15.0), 278.2196 (10.1)	N.A.	PE
78	43.5	293.2111	C_18_H_28_O_3_	0.0	219.1745 (100), 275.2003 (16.6), 220.1779 (14.4)	N.A.	PE
79	43.8	372.3458	C_22_H_45_NO_3_	3.9	300.2881 (100), 282.2777 (59.6), 372.3455 (57.9), 301.2915 (18.7), 373.3492 (13.0), 283.2811 (10.1)	N-[2-Hydroxy-1-(hydroxymethyl)heptadecyl]butanamide	NBA
80	44.0	279.2324	C_18_H_30_O_2_	−0.6	279.2324 (100), 123.1181 (88.3), 137.1335 (77.6), 173.1331 (43.3), 135.1179 (33.3), 209.1542 (26.5)	Linolenic acid	PE, TCM
81	44.1	300.2894	C_18_H_37_NO_2_	0.3	282.2791 (100), 265.2525 (24.9), 247.2418 (23.2), 283.2820 (15.8)	N.A.	Water
82	44.2	372.3478	C_22_H_45_NO_3_	−1.4	328.3213 (100), 310.3107 (65.9), 372.3473 (58.3), 329.3243 (18.3), 373.3505 (12.7), 311.3136 (11.1)	N-[2-Hydroxy-1-(hydroxymethyl)eicosyl]acetamide	EA
83	44.6	295.2264	C_18_H_30_O_3_	1.4	151.1114 (100), 277.2162 (82.7), 249.2212 (23.0), 133.1005 (14.5), 161.1321 (14.0)	FA 18:3+1O	PE, TCM
84	44.7	590.4619	C_32_H_63_NO_8_	2.6	590.4617 (100), 300.2888 (75.9), 462.3417 (69.6), 222.0965 (36.7), 428.4089 (31.4), 282.2783 (22.1)	N-1-[(β-d-Glucopyranosyloxy) methyl]-2-hydroxyheptadecyl] octanamide	NBA
85	45.1	279.1604	C_16_H_22_O_4_	−1.3	149.0246 (100), 167.0347 (1.2)	Dibutyl phthalate	PE
86	45.1	743.4219	C_38_H_62_O_14_	−0.9	383.2051 (82.6), 361.2231 (37.5), 185.0814, 129.0188	N.A.	TCM, EA
87	45.4	400.3765	C_24_H_49_NO_3_	5.0	328.3193 (100), 400.3767 (63.0), 310.3087 (57.7), 329.3226 (21.8)	N-[2-Hydroxy-1-(hydroxymethyl)eicosyl]butanamide	NBA
88	45.6	324.2903	C_20_H_37_NO_2_	−0.6	324.2895 (100), 179.1070 (65.5), 307.2647 (48.0), 263.2367 (25.0)	Linoleoyl ethanolamide	PE, TCM
89	46.2	403.2320	C_20_H_34_O_8_	1.6	185.0804 (100), 157.0127 (38.0), 129.0180 (37.7), 259.1532 (22.6), 139.0023 (14.7), 217.0334 (10.9)	Acetyl tributyl citrate	EA, NBA, Water
90	46.2	425.2144	C_17_H_32_N_2_O_10_	−3.5	425.2142 (100), 426.2172 (20.9), 365.1931 (14.4)	N.A.	TCM
91	46.7	429.3004	C_27_H_40_O_4_	−0.5	429.3002 (100), 411.2890 (34.7), 271.2051 (11.6)	9(11)-dehydrohecogenin	PE
92	46.8	607.2536	C_34_H_38_O_10_	0.2	607.2537 (100), 608.2537 (38.8), 609.2640 (13.6)	N.A.	TCM
93	47.2	428.4093	C_26_H_53_NO_3_	1.3	300.2887 (100), 282.2779 (42.4), 428.4084 (36.3), 301.2922 (19.7)	N-[2-Hydroxy-1-(hydroxymethyl)heptadecyl]octanamide	NBA
94	47.7	323.2584	C_20_H_34_O_3_	−0.3	277.2158 (100), 151.1121 (59.7), 179.1430 (43.3), 135.1174 (16.1), 161.1329 (14.1)	N.A.	PE
95	47.9	609.2691	C_34_H_40_O_10_	0.5	609.2687 (100), 610.2719 (39.2)	N.A.	TCM, EA, NBA

PE, petroleum ether; TCM, chloroform; EA, ethyl acetate; NBA, n−butanol.

**Table 2 molecules-27-02724-t002:** Cytotoxicities of different fractions from PPL.

Extraction of Parts	IC_50_ (μg/mL)				
	A549	MCF-7	HepG2	A431	HBE
Petroleum ether	6.12 × 10^1^	1.50 × 10^2^	3.29 × 10^1^	1.32 × 10^2^	1.78 × 10^2^
Chloroform	1.47 × 10^1^	1.26 × 10^1^	1.36 × 10^1^	1.43 × 10^1^	1.52 × 10^2^
Ethyl acetate	4.99 × 10^2^	2.87 × 10^2^	1.32 × 10^2^	1.45 × 10^2^	2.82 × 10^2^
n-Butanol	1.41 × 10^1^	1.18 × 10^1^	0.910 × 10^1^	1.68 × 10^1^	1.09 × 10^2^
Water	8.30 × 10^3^	7.30 × 10^3^	7.60 × 10^3^	6.20 × 10^3^	−
Cisplatin	0.512 × 10^1^	0.664 × 10^1^	0.339 × 10^1^	0.361 × 10^1^	4.35 × 10^1^

**Table 3 molecules-27-02724-t003:** The main variables identified based on VIP > 1 and *p*-value < 0.05 between HA and LA.

Var ID	RT (min)	Compound	VIP	*p*-Value	HA	LA
66	40.3	1-*O*-(β-d-glc)-(2*S*, 3*S*)-2-acetamide-4 (*E*)-octadecane-1,3,6-triol	4.76	1.80 × 10^−14^	*	
62	39.5	Hydroxyl 1-*O*-(β-d-glc)-(2*S*, 3*S*)-2-acetamide-4 (*E*)-octadecane-1,3,6-triol	3.81	8.59 × 10^−19^	*	
83	44.6	FA 18:3+1O	3.49	4.33 × 10^−2^		*
74	42.5	N.A.	3.21	3.76 × 10^−4^	*	
72	42.0	1-Deoxy-1-(tetradecylamino)-d-fructose	3.19	1.12 × 10^−4^	*	
22	15.9	Loliolide	3.03	8.79 × 10^−4^	*	
45	27.4	Pariphyllin A	2.94	1.03 × 10^−2^	*	
69	41.3	Phytosphingosine	2.81	1.16 × 10^−11^	*	
76	43.0	1-[Bis(2-Hydroxyethyl)Amino]Hexadecan-2-*O*l	2.77	2.01 × 10^−2^		*
84	44.7	N-[1-[(β-d-Glucopyranosyloxy) methyl]-2-hydroxyheptadecyl] octanamide	2.75	4.58 × 10^−4^	*	
87	45.4	N.A.	2.60	4.16 × 10^−4^	*	
47	28.2	Pseudoproto-gracillin	2.58	1.96 × 10^−2^	*	
80	44.0	Linolenic acid	2.38	3.95 × 10^−2^		*
75	42.8	d-erythro-N-Acetylsphinganine	2.37	1.61 × 10^−2^	*	
93	47.2	N-[2-Hydroxy-1-(hydroxymethyl)heptadecyl]octanamide	2.36	4.71 × 10^−4^	*	
95	47.9	N.A.	2.30	2.04 × 10^−3^	*	
64	40.0	Chonglouside H	2.26	9.81 × 10^−20^	*	
21	15.6	Crustecdysone	2.12	3.49 × 10^−3^	*	
46	27.7	Pseudoproto-Pb	1.99	1.72 × 10^−2^	*	
20	14.9	Kaempferol 7-*O*-neohesperidoside	1.95	1.69 × 10^−3^	*	
18	13.8	Luteolin-3’-*O*-Glc-(l→2)-glucoside	1.93	1.29 × 10^−3^	*	
68	41.2	N-tetradecyl-d-gluconamide	1.91	1.70 × 10^−6^	*	
25	17.8	Kaempferol 3-Glucoside	1.84	1.26 × 10^−2^	*	
49	29.9	27-*O*-glc-(25*R*)-spirost-5-ene-3β,27-diol-3-*O*-rha(1→4)-[rha(1→2)]-glc or its isomer	1.82	7.30 × 10^−3^	*	
54	33.0	Chonglouoside SL-14 or its isomer	1.80	4.18 × 10^−3^	*	
60	38.4	HexCer t18:0	1.79	2.31 × 10^−23^	*	
53	32.6	27-*O*-glc-(25*R*)-spirost-5-ene-3β,27-diol-3-*O*-rha(1→4)-[rha(1→2)]-glc or its isomer	1.70	9.45 × 10^−4^	*	
36	22.9	Th	1.56	2.70 × 10^−2^	*	
8	11.0	Luteolin-3’,7-di-*O*-glucoside	1.34	4.92 × 10^−2^	*	
59	37.5	(25*S*)-spirost-5-ene-3β,25-diol-3-*O*-rha (1→2)-[glc (1→3)] -glc	1.16	4.34 × 10^−5^	*	

RT, retention time; VIP, variable importance in the projection; HA, high-activity group; LA, low-activity group. “*” indicates where the compound is present.

## Data Availability

Data are contained within the article.

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
