# Peer review of "Study of Chemical Compositions and Anticancer Effects of Paris polyphylla var. Chinensis Leaves"

_molecules, 2022, doi:10.3390/molecules27092724_

Round 1

Reviewer 1 Report

molecules-1616089

Title: Study of chemical compositions and anti-cancer effects of Paris polyphylla leaves

In this paper, authors isolated the major composition of different fractions from the Paris polyphylla leaves (PPL). Meanwhile, the toxicity of different fractions to human lung cancer A549 cells was tested by MTT assay. This paper did not show clear evidence the function role of PPL extracts in A549 cells. To improve the manuscript, some additional experiments are needed.

Major revision

  1. Authors compared the cytotoxicity of these compounds against normal cells.
  2. Why did you only tested cytotoxicity against A549 cells? Authors should be compared the cancer cells-specificity or selectivity using various cancer cell lines.
  3. Authors should be compared the cytotoxicity of these compounds with chemotherapeutic drugs.
  4. What is the major anticancer activity of these components?

Author Response

Response to Reviewer 1 Comments

Dear reviewer,

  We would like to thank you for the careful reading of our manuscript (molecules-1616089) entitled “Study of chemical compositions and anti-cancer effects of Paris polyphylla leaves”. According to the comments, we have carefully revised the manuscript and found these comments are helpful in improving the quality of the manuscript. Listed below are the reviewers' comments and the changes that were made in the manuscript.

This manuscript investigated the “Study of chemical compositions and anti-cancer effects of Paris polyphylla leaves”. The following comments have been outlined to improve the quality.

Question 1: Authors compared the cytotoxicity of these compounds against normal cells.

Answer 1: Thanks for the review’s comment. On the one hand, as a traditional Chinese medicine, Paris polyphylla has a long history of clinical application in China and its safety has been verified. On the other hand, the metabolism of tumor cells and normal cells is very different, so it’s impossible to compare the cytotoxicity of these compounds by direct comparison of these two cells. In this work, the main goal is to discover the toxicity difference of different polar fractions on A549 cancer cells, and screen all compounds that may produce activity through multivariate statistical analysis, while the safety of the drug will be preliminarily explored in the later specialized toxicological study.  

Question 2: Why did you only tested cytotoxicity against A549 cells? Authors should be compared the cancer cells-specificity or selectivity using various cancer cell lines.

Answer 2: Thanks for the review’s comment. It has been reported that the extract of Paris polyphylla leaves pronounced inhibitory activity on the proliferation of A549 cancer cells. It also induced apoptosis in A549 cells, most probably by a mechanism related to Ki-67 and p21 ras protein expression, and arrest of cell cycle in G0/G1-phase(Hu, et al. 2017). However, there is no report on the pharmacodynamic substance basis of the extract of Paris polyphylla leaves against A549 cancer cells, so this study was carried out on this basis. We would emphasize this purpose in the Introduction paragraph in line

   The main purpose of this work is to screen the pharmacoactive substances in the leaves of Paris polyphylla against A549 cancer cells. However, the title of this article is not appropriate and may cause confusion to readers. So, the title “Study of chemical compositions and anti-cancer effects of Paris polyphylla leaves” was changed to “Identification of chemical constituents of Paris polyphylla leaves and anti-cancer effects in A549 lung cancer cells”.

Question 3: Authors should be compared the cytotoxicity of these compounds with chemotherapeutic drugs.

Answer 3: Thanks for the review’s comment. Lack of chemotherapeutic drugs as positive controls may be a deficiency in our experimental design. However, in our experiments, we used the mixtures of different extraction conditions. It is difficult for us to make accurate qualitative and quantitative analysis of these mixtures. Our experiment was designed to study whether the extracts had inhibitory effect on A549 cancer cells. And the degree of inhibition and specific active compounds are issues to be considered in our follow-up work.

Question 4: What is the major anticancer activity of these components?

Answer 4: Thanks for the review’s comment. In this study, it was found that chloroform and n-butanol fractions had the best inhibitory effect on A549 cancer cells, and the proportion and content of steroid saponins, flavonoids and ceramides in these fractions were higher, and these substances might be the active components of anti-A549 cancer cells. We would emphasize this conclusion in the Results paragraph and make relevant changes in the Results paragraph.

Hu R, Yu W, Zhuo Y, Yang Y, Hu X. 2017. Paris polyphylla extract inhibits proliferation and promotes apoptosis in A549 lung cancer cells. Tropical Journal of Pharmaceutical Research 16.

Reviewer 2 Report

I revised the submission with the title "Study of chemical compositions and anti-cancer effects of Paris polyphylla leaves"

In my view the article still needs a major revision to make it clearer for the reader. 

More details need to be given about the methodology, LC-MS and Botanic terms, and incorporated into the text. For example:

1. Was any chemical voucher deposited in any herbarium? Please provide it.

2. What method was used to obtain the fractions? Liquid-liquid extraction? 

3. What was the parameters used to process the LC-MS data? Which software? Number of features detected using the given parameters? What is the scale (Pareto or UV) of the data used to perform the multivariate analysis (PCA)? Please give more details of the processing methods of LC-MS data.

4. The legend of all figures need to be more detailed. 
For example, figure 1 shows the chromatograms in which polarity, positive or negative?

5. A major revision should be made in the text for italicization (i.e. m/z, plant name...). 

6. Abbreviations should be written in full the first time they appear in each section of the article.

7. In Figure 3, is this fragmentation route already described in the literature or is it proposed in this paper by the authors? Please specify in other figures also.

8. The authors mention in lines 272-274 the preparation of quality control samples but no data is presented afterwards. 
What was the strategy for using the QC samples? Were they injected at regular intervals between the test samples? This strategy needs to be detailed and the results (analysis) presented in the text, including the PCA.

9. In the following sentence "flavonoids and ceramides, were highly expressed in HA and were the key active substances for inhibiting A549 cancer cells." 

I don't think you can claim that they are the active compounds since they have not been tested. The study may suggest that they are responsible. You also need to clarify some terms in this sentence but which appear elsewhere in the text, for example "expressed". Since no specific analysis was done to quantify the compounds, it must be clear in the text that you are comparing the relative intensity of the compounds in the different fractions. This should be revised throughout the text. 

9. Line 26 - "The Paris polyphylla (PP) is a perennial herb of genus Paris (Trilliaceae), distributed". This need to be rephrased. for i.e. "The Paris polyphylla (PP) is a perennial herb belonging to Trilliaceae family..." Please cite also the plant name author (botanical authority who first describe the plant).

10. Figures 6 and 7 should be merged. 

11. The term "polar" should be used in a comprehensible and concise form. For exemple:  "Figure 1. Base Peak chromatogram (BPC) of different polar fractions.(A). Petroleum ether fraction; (B). Chloroform frac- 83 tion; (C). Ethyl acetate fraction; (D). n-Butanol fraction; (E). Water fraction."  Nonetheless, chloroform is a non-polar solvent. It should be revised also in the text.

Author Response

Response to Reviewer 2 Comments

Dear reviewer,

  We would like to thank you for the careful reading of our manuscript (molecules-1616089) entitled “Study of chemical compositions and anti-cancer effects of Paris polyphylla leaves”. According to the comments, we have carefully revised the manuscript and found these comments are helpful in improving the quality of the manuscript. Listed below are the reviewers' comments and the changes that were made in the manuscript.

This manuscript investigated the “Study of chemical compositions and anti-cancer effects of Paris polyphylla leaves”. The following comments have been outlined to improve the quality.

Question 1: Was any chemical voucher deposited in any herbarium? Please provide it.

Answer: Thanks for the review’s comment. The chemical voucher of Paris polyphylla leaves was supplemented in the Materials and Methods paragraph.

  “Voucher specimens (No. PPL 20200923) were deposited in Moganshan Campus of Zhejiang University of Technology.” were added in line 295-296.

Question 2: What method was used to obtain the fractions? Liquid-liquid extraction? 

Answer: Thanks for the review’s comment. These fractions were obtained by liquid-liquid extraction. We emphasized this extract method in the Materials and Methods paragraph. 

  “The residue was suspended in distilled water and successively extracted with different polar solvents including petroleum ether, chloroform, ethyl acetate and n-butanol.” was modified to “The residue was suspended in distilled water and successively extracted with different polar solvents including petroleum ether, chloroform, ethyl acetate and n-butanol by liquid-liquid extraction.” in line 313.

Question 3: What was the parameters used to process the LC-MS data? Which software? Number of features detected using the given parameters? What is the scale (Pareto or UV) of the data used to perform the multivariate analysis (PCA)? Please give more details of the processing methods of LC-MS data.

Answer: Thanks for the review’s comment. It was an oversight of ours not to describe the processing methods of LC-MS data in detail, and the detailed process of LC-MS data processing is as follows “The MS/MS data was collected by Hystar 3.2 software and the collected data of six batches were internally corrected in enhanced mode by Bruker Daltonics DataAnalysis 4.4 software with sodium formate calibration solution. The processed data was converted into Analysis Base File (ABF) format using AbfConverter (Version 4.0.0), and then imported into MS-Dial (Version 4.16) for pre-processing including peak collection, peak discrimination, deconvolution, filtering, peak alignment and normalization. After that, a data matrix containing average retention time, average retention index, EI spectrum, sample name and peak area was obtained. Then, the three-dimensional data matrix was introduced into SMICA 14.1 (Umetrics, Sweden) for PCA and OPLS-DA. All of the variables were normalized by Par (Pareto-scaled) before stoichiometric analysis. ” in line 357-367.

Question 4: The legend of all figures need to be more detailed. 

For example, figure 1 shows the chromatograms in which polarity, positive or negative?

Answer:Thank you for the careful reading of our manuscript. The legend of figures have been revised.

  • . “Figure 1.Base Peak chromatogram (BPC) of different polar fractions.(A). Petroleum ether fraction; (B). Chloroform fraction; (C). Ethyl acetate fraction; (D). n-Butanol fraction; (E). Water fraction.” was changed to “Figure 1. Base Peak chromatogram (BPC) of different fractions from the Paris polyphylla leaves (PPL) in positive ion mode.(A). Petroleum ether fraction; (B). Chloroform fraction; (C). Ethyl acetate fraction; (D). n-Butanol fraction; (E). Water fraction.” in line 107-108.
  • . “Table Compounds identified in the extracts of different fractions by UPLC-Q-TOF-MS/MS.” was changed to “Table 1. Compounds identified in the extracts of different fractions from PPL by UPLC-Q-TOF-MS/MS.” in line 114.
  • “PE, petroleum ether; TCM, chloroform; EA, ethyl acetate; NBA, n-butanol”was add below Table 1 in line 
  • Figure 2.The steroidal saponins skeleton identified from different parts of PPL” was changed to “Figure 2. The steroidal saponins skeleton identified from different parts of PPL in positive ion mode.” in line 126-127.
  • . “Figure 3.The fragmentation pathways of compound 44 .” was changed to “Figure 3. The fragmentation pathways of compound 44 in positive ion mode” in line 141.
  • . “Figure 4.The fragmentation pathways of compound 64, 40, 36.” was changed to “Figure 4. The fragmentation pathways of compound 64, 40, 36 in positive ion mode.” in line 185.
  • . “Figure 5.The flavonoids skeleton identified from different parts of PPL.” was changed to “Figure 5. The flavonoids skeleton identified from different parts of PPL in positive mode.” in line 194.
  • . “Table 2. In vitro growth inhibition of different fractions determined in A549 cell lines.”was changed to “Table 2. In vitro growth inhibition of different fractions from PPL determined in A549 cancer cell lines.” in line 246.
  • . “Figure 6.Score plots from principal components analysis (PCA).” and “Figure 7. Orthogonsl partial least squares discriminant analysis (A) and loading plot (B) models in different polar layers.” were changed to “Figure 6. Score plots from principal components analysis (A), orthogonsl partial least squares discriminant analysis (B) and loading plot models (C) in different layers from PPL.” in line 270-271.
  • . In table 3, “HA: high activity group LA: low activity group.”was changed to “RT, retention time; VIP, Variable Importance in the Projection; HA, high activity group; LA, low activity group.” was added in line 274.

Question 5: A major revision should be made in the text for italicization (i.e. m/z, plant name...). 

Answer: Thank you for the careful reading of our manuscript. The italics in the text have been modified.

  • . “Paris polyphylla”was changed to “Paris polyphylla” in lines 3, 25, 27, 36, 46, 48, 49, 69, 107.
  • “m/z”was changed to “m/z” in lines 131, 133-135, 148-151, 155, 158, 161-162, 173, 175, 177-179, 197, 199, 201, 206-207, 211-212, 220, 225-227.
  • . “p-value”was changed to “p-value” in line 262.

Question 6: Abbreviations should be written in full the first time they appear in each section of the article.

Answer: Thanks for the review’s comment. The abbreviations have been written in full the first time they appear.

  • . “ polyphylla var. Chinese and P. polyphylla var. Yunnanensis”was changed to “Paris polyphylla var. chinensis and Paris polyphylla var. yunnanensis” in line 49.
  • . “ The components of different polar fractions were analyzed by UPLC/Q-TOF MS with MS/MS”was changed to “The components of different fractions were analyzed by ultra performance liquid chromatography quadrupole time-of-flight mass spectrometry with MS/MS (UPLC/Q-TOF MS/MS)” in line 80-82.

Question 7: In Figure 3, is this fragmentation route already described in the literature or is it proposed in this paper by the authors? Please specify in other figures also.

Answer: The fragmentation routes of compound 44, 64, 40 and 36 were proposed based on the fragment information in this paper. The description were modified as follows:

  • . “The fragmentation pathways of compound 44 wasshowed in Figure 3.” was  changed to “The fragmentation pathways of compound 44 was proposed in Figure 3.” in line 138.
  • . “The fragmentation pathways of compound 64 wasshowed in Figure 4. ” was  changed to “The fragmentation pathways of compound 64 was proposed in Figure 4. ” in line 153.
  • . “The fragmentation pathways of compound 40was showed in Figure 4. ” was  changed to “The fragmentation pathways of compound 40 was proposed in Figure 4. ” in line153.
  • . “The fragmentation pathways of compound 36was showed in Figure 4. ” was  changed to “The fragmentation pathways of compound 36 was proposed in Figure 4. ” in line 165.

Question 8: The authors mention in lines 272-274 the preparation of quality control samples but no data is presented afterwards. 

What was the strategy for using the QC samples? Were they injected at regular intervals between the test samples? This strategy needs to be detailed and the results (analysis) presented in the text, including the PCA.

Answer: Thanks for the review’s comment. The strategy for using the QC samples was added as described “Each batch of samples was tested 6 times in parallel, and every 6 analyses included an inspection of QC samples. ” in line 339-340, and the results of QC samples in PCA was shown in Figure 6.

Question 9: In the following sentence "flavonoids and ceramides, were highly expressed in HA and were the key active substances for inhibiting A549 cancer cells." 

I don't think you can claim that they are the active compounds since they have not been tested. The study may suggest that they are responsible. You also need to clarify some terms in this sentence but which appear elsewhere in the text, for example "expressed". Since no specific analysis was done to quantify the compounds, it must be clear in the text that you are comparing the relative intensity of the compounds in the different fractions. This should be revised throughout the text. 

Answer: This expression has been modified in the text.

  • . “As shown in Table 3, 30 compounds were screened, of which 27 compounds, including steroidal saponins, flavonoids and ceramides, were highly expressed in HA and were the key active substances for inhibiting A549 cancer cells.”was changed to “As shown in Table 3, 30 compounds were screened, of which 27 compounds, including steroidal saponins, flavonoids and ceramides were highly contained in HA,  which might be responsible for inhibiting A549 cancer cells.” in line 265.
  • . “The results of MTT test and multivariate statistical analysis showed that the steroidal saponins, flavonoids and ceramides mainly contained in chloroform and n-butanol fractions may be the active components of PPL inhibiting A549. ”was changed to “The results of MTT test and multivariate statistical analysis showed that the proportions and contents of steroidal saponins, flavonoids and ceramides in chloroform and n-butanol fractions were higher, which might be the reason for the high inhibitory effect on A549 cancer cells.” in line 371-374.

Question 10: Line 26 - "The Paris polyphylla (PP) is a perennial herb of genus Paris (Trilliaceae), distributed". This need to be rephrased. for i.e. "The Paris polyphylla (PP) is a perennial herb belonging to Trilliaceae family..." Please cite also the plant name author (botanical authority who first describe the plant).

Answer: Thanks for the review’s comment. The Paris polyphylla was first recorded by Saoxiu in Shennong Ben Cao Jing in China, while the botanical authority who first describe the plant in English has not been found.

 “The Paris polyphylla (PP) is a perennial herb of genus Paris (Trilliaceae), distributed in southwest China.” was changed to “The Paris polyphylla (PP) which was first recorded by Saoxiu in Shennong Ben Cao Jing in China is a perennial herb belonging to the Trilliaceae family, which is distributed in the southwest China.” in line 46-48.

Question 11: Figures 6 and 7 should be merged. 

Answer: Figure 6 and 7 have been merged in Figure 6.

Question 12: The term "polar" should be used in a comprehensible and concise form. For exemple: "Figure 1. Base Peak chromatogram (BPC) of different polar fractions.(A). Petroleum ether fraction; (B). Chloroform fraction; (C). Ethyl acetate fraction; (D). n-Butanol fraction; (E). Water fraction."  Nonetheless, chloroform is a non-polar solvent. It should be revised also in the text.

Answer: Since chloroform is a non-polar solvent, and the description of solvent polarity is not appropriate in this paper. These issues have been revised in the text.  

  • . “ In the study, solvents of different polarities and sequential extraction were used for PPL.”was changed to “In the study, solvents of different types and sequential extraction were used for PPL.” in line 79.
  • . “The components of different polar fractions were analyzed by ultra performance liquid chromatography quadrupole time-of-flight mass spectrometry with MS/MS(UPLC/Q-TOF MS with MS/MS)”was changed to “The components of different fractions were analyzed by ultra performance liquid chromatography quadrupole time-of-flight mass spectrometry with MS/MS(UPLC/Q-TOF MS with MS/MS)” in line 80.
  • . “Identification of the chemical composition from different polar fractions”was changed to “Identification of the chemical composition from different fractions” in line 86.
  • . “Figure 1.Base Peak chromatogram (BPC) of different polar fractions from the Paris polyphylla leaves (PPL) in positive ion mode. ” was changed to “Figure 1. Base Peak chromatogram (BPC) of different fractions from the Paris polyphylla leaves (PPL) in positive ion mode.”in line 107.
  • . “Table 1.Compounds identified in the extracts of different polar fractions from PPL by UPLC-Q-TOF-MS/MS.”was changed to “Table 1.Compounds identified in the extracts of different fractions from PPL by UPLC-Q-TOF-MS/MS.” in line 114.
  • . “2.2 Antitumor activity of different polar fractions in vitro”was changed to “2.2 Antitumor activity of different fractions in vitro” in line 237.
  • . “Table 2.In vitro growth inhibition of different polar fractions from PPL determined in A549 cancer cell lines.” was changed to “Table 2. In vitro growth inhibition of different fractions from PPL determined in A549 cancer cell lines.” in line 246.
  • . “The residue was suspended in distilled water and successively extracted with different polar solvents including petroleum ether, chloroform, ethyl acetate and n-butanol by liquid-liquid extraction.”was changed to “The residue was suspended in distilled water and successively extracted with different solvents including petroleum ether, chloroform, ethyl acetate and n-butanol by liquid-liquid extraction.” in line 312.
  • . “In the study, UPLC/Q-TOF MS/MS was used to qualitatively analyze the chemical conposition of different polar fractions in PPL alcohol extracts and 79 compounds were finally identified. ”was changed to “In the study, UPLC/Q-TOF MS/MS was used to qualitatively analyze the chemical conposition of different fractions in PPL alcohol extracts and 79 compounds were finally identified.” in line 370.

Round 2

Reviewer 1 Report

This paper did not clearly revise according to the reviewer's comments.

Authors must be answered the comments number 1 and 2. In addition, added data on the new experiments.

Author Response

Response to Reviewer 1 Comments

Dear reviewer,

We would like to thank you for the careful reading of our manuscript (molecules-1616089) entitled “Study of chemical compositions and anti-cancer effects of Paris polyphylla leaves”. According to the comments, we have carefully revised the manuscript and found these comments are helpful in improving the quality of the manuscript. Listed below are the reviewers' comments and the changes that were made in the manuscript.

This manuscript investigated the “Study of chemical compositions and anti-cancer effects of Paris polyphylla leaves”. The following comments have been outlined to improve the quality.

Question : This paper did not clearly revise according to the reviewer's comments.

Authors must be answered the comments number 1 and 2. In addition, added data on the new experiments.

Comment number 1 : Authors compared the cytotoxicity of these compounds against normal cells.

Comment number 2 : Why did you only tested cytotoxicity against A549 cells? Authors should be compared the cancer cells-specificity or selectivity using various cancer cell lines.

Answer: We have added three sets of cancer cell lines (MCF-7, HepG2 and A431) for cytotoxicity assays and a cisplatin-positive set. The specific experimental data are shown in Table 2. The results showed that both chloroform and n-butanol fractions had strong inhibitory effects on these four cancer cells, and were less toxic to human normal cells. The full text has been revised accordingly.

Table 2. Cytotoxicity of different fractions from PPL against four human cancer cells.

Extraction of parts

IC50 (μg/mL)

A549

MCF-7

HepG2

A431

HBE

Petroleum ether 

6.12×101

1.50×102

3.29×101

1.32×102

1.78×102

 Chloroform

1.47×101

1.26×101

1.36×101

1.43×101

1.52×102

Ethyl acetate

4.99×102

2.87×102

1.32×102

1.45×102

2.82×102

n-Butanol

1.41×101

1.18×101

0.910×101

1.68×101

1.09×102

Water

8.30×103

7.30×103

7.60×103

6.20×103

-

Cisplatin

0.512×101

0.664×101

0.339×101

0.361×101

4.35×101

  • “Meanwhile, the toxicity of different fractions to human lung cancer A549 cells was tested by MTT assay.”was changed to “Meanwhile, the cytotoxic activity of the extracts of different fractions against four selected human cancer cell lines and one human .” in line 15-17.
  • “High levels of steroidal saponins and flavonoids observed from n-butanol extracts were strongly inhibitory to A549 cells.”was changed to “Finally, more than 60 compounds were obtained and identified from different fractions of Paris polyphylla chinensis leaves, and the chloroform and n-butanol extracts showed significant cytotoxic effects on these four cancer cells.” in line 18-21.
  • The key words were changed to “Paris polyphylla chinensis; leaves; identification; cytotoxic effect ”in line 26.
  • “Therefore, more detailed studies on the phytochemical compositions and pharmacological effects to screen the pharmacoactive substances in the PPL against A549 cancer cells are of great interest.”was changed to “Therefore, it is of great significance to conduct more detailed studies on phytochemical components and pharmacological effects to screen the pharmacoactive substances in PPL .” in line 56-58.
  • “The differences in the toxicity of different extracts to A549 cells were studied by MTT method.”was changed to “The cytotoxicity differences of different extracts were studied by MTT method.” in line 65-66.
  • The Antitumor activity of different fractions in vitro paragraphwas changed to “All fractions were evaluated for their cytotoxic activities against four human cancer cell lines and one human normal cell line, including A549 (human lung carcinoma), MCF-7 (human breast carcinoma), HepG2 (human hepatoellular carcinomas), A431 (epidermoid carcinoma cell) and HBE (human bronchial epithelial cell). The results were shown in Table 2. Compared with the positive control cisplatin, these fractions have certain cytotoxicity and are less toxic to human normal epithelial cells. Among these fractions, the n-butanol and chloroform extracts showed the most obvious inhibitory effects on these four cancer cells. The n-butanol extract strongly inhibited the proliferation of HepG2 cells with an IC50 value of 0.910×101 μg/mL, and the chloroform extracts also exhibited considerable cytotoxicity to MCF-7 cells with an IC50 value of 1.26×101 μg/mL. Other fractions had moderate or weak cytotoxicity with IC50 values ranging from 3.29×101 to 8.30×103 μg/mL.” in line 211-222.
  • The title of Table 2was changed to “Table 2. Cytotoxicity of different fractions from PPL.” in line 225.
  • “To differentiate the potential substance basis of PPL on inhibiting A549 cancer cells, metabolites were analyzed using multivariate statistical methods. ”was changed to “To differentiate the potential substance basis for PPL inhibition of these cancer cells, metabolites were analyzed using multivariate statistical methods.” in line 227.
  • “As shown in Table 3, 30 compounds were screened, of which 27 compounds, including steroidal saponins, flavonoids and ceramides, were highly contained in HA which might be responsible for inhibiting A549 cancer cells.”was changed to “As shown in Table 3, 30 compounds were screened, of which 27 compounds, including steroidal saponins, flavonoids and ceramides, were more abundant in high activity group.” in line 240-242.
  • The literature about the anticancer activities of Pariphyllin A, pseudoproto-gracillin, pseudoproto-Pb, Chonglouside 14 and ceramides were added to enrich the discussion of multivariate statistical analysisin line 248-279. 
  • “A549 cells were purchased from Hunan Fenghui Biotechnology Co. Ltd. Roswell Park Memorial Institute (RPMI-1640) and Fetal Bovine Serum (FBS) were obtained from Gbico (New York, USA).”was changed to “A549 cells, MCF-7 cells, HepG2 cells, A431 cells and HBE cells were purchased from Hunan Fenghui Biotechnology Co. Ltd. Roswell Park Memorial Institute (RPMI-1640), Dulbecco's Modified Eagle Medium (DMEM) and Fetal Bovine Serum (FBS) were obtained from Gbico (New York, USA).” in line 294-297.
  • “Human lung cancer A549 cells were cultured in RPMI-1640 medium, supplemented with 10% fetal bovine serum and cultured in a cell incubator at 37°C and 5% CO2. A suspension of the A549 cells at logarithmic phase was seeded in 96-well plates at a density of 5000 cells per well (100 μL of medium per well) and cultured 24h for cell adhesion.”was changed to “Four human cancer cell lines (A549, MCF-7, HepG2, and A431) and one human normal epithelial cell line were used for cytotoxicity assays. All cells were cultured in RPMI-1640 or DMEM medium, supplemented with 10% fetal bovine serum at 37°C and 5% CO2 in a cell incubator. ” in line 332-335.
  • “Each cell line was exposed to different concentrations of test samples for 48 h, with cisplatin as a positive control. ”was added in line 339-340.
  • The Conclusions was changed to “The results of MTT assay and multivariate statistical analysis showed that the chloroform and n-butanol fractions exhibited interesting cytotoxicity against these four human cancer cell lines, with IC50values ranging from 0.910×101 to 1.68×101 μg/mL, less toxic to human normal epithelial cell line. The ratio and content of steroidal saponins, flavonoids and ceramides were higher in the chloroform and n-butanol fractions, which may account for the high inhibitory effect on cancer cells.” in line 363-368.

Reviewer 2 Report

The authors have made an effort and answered the questions. However, some minor corrections are still needed in my view.

The plant names should be revised following the “international plant names index” as «Paris polyphylla Sm.” “Paris polyphylla var. chinensis (Franch.) H.Hara » and « Paris polyphylla var. yunnanensis(Franch.) Hand.-Mazz ».

About the identification of the mass fragments that the authors propose, even if they are common fragments, I believe it is the best practice to reference them. The literature is vast in describing such fragmentation pathways. Please include those most pertinent references to your work. 

Author Response

Response to Reviewer 2 Comments

Dear reviewer,

We would like to thank you for the careful reading of our manuscript (molecules-1616089) entitled “Study of chemical compositions and anti-cancer effects of Paris polyphylla leaves”. According to the comments, we have carefully revised the manuscript and found these comments are helpful in improving the quality of the manuscript. Listed below are the reviewers' comments and the changes that were made in the manuscript.

This manuscript investigated the “Study of chemical compositions and anti-cancer effects of Paris polyphylla leaves”. The following comments have been outlined to improve the quality.

Question 1: The plant names should be revised following the “international plant names index” as «Paris polyphylla Sm.” “Paris polyphylla var. chinensis (Franch.) H.Hara » and « Paris polyphylla var. yunnanensis(Franch.) Hand.-Mazz ».

Answer: The plant names were revised according to the “international plant names index”. 

  • Paris polyphylla”was changed to “Paris polyphlla  chinensis” in title and line 15, 20, 26 and 54.
  • Paris polyphylla”was changed to “Paris polyphlla  chinensis (Frand.) Hara” in line 12.
  • Paris polyphylla chinensisand Paris polyphylla var. yunnanensis” was changed to “Paris polyphlla var. chinensis (Frand.) Hara and Paris polyphylla var. yunnanensis (Franch.) Hand.-Mazz.” in line 32-33.

 Question 2: About the identification of the mass fragments that the authors propose, even if they are common fragments, I believe it is the best practice to reference them. The literature is vast in describing such fragmentation pathways. Please include those most pertinent references to your work. 

Answer: Thank you for reading our manuscript carefully. References to the identification of the mass fragments were added in line 117 and 160. These literatures summarize the fragmentation behavior of steroid saponins and are of great help to the identification of mass fragments in this paper.  

Round 3

Reviewer 1 Report

This paper clearly revised according to the reviewer's comments

Reviewer 2 Report

The authors answered all my questions satisfactorily.